# Interpretable Machine Learning for Personalized Medical Recommendations: A LIME-Based Approach

**DOI:** 10.3390/diagnostics13162681

**Published:** 2023-08-15

**Authors:** Yuanyuan Wu, Linfei Zhang, Uzair Aslam Bhatti, Mengxing Huang

**Affiliations:** School of Information and Communication Engineering, Hainan University, Haikou 570100, China; wyuanyuan82@163.com (Y.W.); 21210810000039@hainanu.edu.cn (L.Z.)

**Keywords:** medical recommendation system, LIME, RF algorithm, gradient boosting, deep learning

## Abstract

Chronic diseases are increasingly major threats to older persons, seriously affecting their physical health and well-being. Hospitals have accumulated a wealth of health-related data, including patients’ test reports, treatment histories, and diagnostic records, to better understand patients’ health, safety, and disease progression. Extracting relevant information from this data enables physicians to provide personalized patient-treatment recommendations. While collaborative filtering techniques and classical algorithms such as naive Bayes, logistic regression, and decision trees have had notable success in health-recommendation systems, most current systems primarily inform users of their likely preferences without providing explanations. This paper proposes an approach of deep learning with a local interpretable model–agnostic explanations (LIME)-based interpretable recommendation system to solve this problem. Specifically, we apply the proposed approach to two chronic diseases common in older adults: heart disease and diabetes. After data preprocessing, we use six deep-learning algorithms to form interpretations. In the heart-disease data set, the actual model recommendation of multi-layer perceptron and gradient-boosting algorithm differs from the local model’s recommendation of LIME, which can be used as its approximate prediction. From the feature importance of these two algorithms, it can be seen that the CholCheck, GenHith, and HighBP features are the most important for predicting heart disease. In the diabetes data set, the actual model predictions of the multi-layer perceptron and logistic-regression algorithm were little different from the local model’s prediction of LIME, which can be used as its approximate recommendation. Moreover, from the feature importance of the two algorithms, it can be seen that the three features of glucose, BMI, and age were the most important for predicting heart disease. Next, LIME is used to determine the importance of each feature that affected the results of the calculated model. Subsequently, we present the contribution coefficients of these features to the final recommendation. By analyzing the impact of different patient characteristics on the recommendations, our proposed system elucidates the underlying reasons behind these recommendations and enhances patient trust. This approach has important implications for medical recommendation systems and encourages informed decision-making in healthcare.

## 1. Introduction

In recent years, the proportion of the elderly in China has increased rapidly, and the elderly themselves are the main population seeking medical treatment and medicine in hospitals. Therefore, in recent years, the health problems of the elderly have gradually attracted wide attention from the public. Various healthcare products for the elderly are becoming popular, but the related healthcare costs are also increasing. These costs are related to conditions that cannot be cured in a short time or that cannot be cured at all. Therefore, to improve the quality of life of the elderly and reduce their disease troubles, experts propose that chronic diseases of the elderly should be prevented. By changing people’s negative behaviors, the incidence of diseases among the elderly can be reduced to improve their living standards. Improvements in negative living habits and specific drug treatments are the most economical approaches. Therefore, in recent years, the recommendation system based on medical care has gradually entered people’s lives and progressively gained in attention, especially in rural and remote mountainous areas with relatively poor medical conditions [1,2]. This can play a more significant role and be more beneficial to the improvement of the living standards of the elderly. For hospitals, chronic diseases of the elderly [3] are difficult to cure, have long treatment periods, and are prone to relapse. The treatment of such diseases tends to be more diversified and requires more attention. As a result, physicians need rich treatment experience, which has led to a shortage of physicians in the field. The healthcare system can make recommendations based on patients’ medical data and current diagnostic reports, which can reduce the level of treatment for chronic diseases among the elderly and enable more elderly people to receive timely treatment from physicians. Therefore, the emergence of the medical recommendation system is expected to help treat chronic diseases affecting the elderly. Of course, medical recommendations cannot completely replace physicians. The system gives certain suggestions to patients and physicians and helps patients to correct negative behaviors in the disease-prevention and -treatment stages. At the same time, using the healthcare system can also reduce treatment times and resources required in emergencies.

The recommendation system was introduced to solve the data explosion. The system is convenient for users to find their own information from among large volumes of data quickly, reducing users’ time consumption. With the rapid development of network information in recent years, the recommendation system plays an increasingly important role in network information. Many effective and professional recommendation methods have appeared and are widely used in e-commerce, news reports, video recommendation [4], social networks, and other fields. Unlike traditional machine learning, recommendation systems can effectively process and analyze data and recommend the results that current users desire most. Although recommendation systems have many advantages, they are not widely used in medicine to take advantage of their potential and benefits. Many methods can make medical recommendations, including collaborative filtering technology [5,6,7], naive Bayes, decision trees, random forest, multi-layer perceptron, and other technologies. While all of these methods can help users find items they might desire, they fail to explain why recommendations are made to users.

In order to solve the problem of unexplainable recommendation systems, LIME-based explainable technology is used in this paper. This system realizes modularization by implementing a LIME-based interpretable recommendation system. This method overcomes the unexplainable problems in the recommendation system and evaluates the recommended medical outcomes based on multiple evaluation indicators. Through the proposed system the following contributions are made:The application performances of different classical algorithms in medical recommendation systems is compared.The interpretability of medical recommendations is realized through the LIME algorithm.

The rest of this article includes the following sections. Section 2 briefly introduces the related work on the medical recommendation system. Section 3 details the proposed recommendation system. Details of the experiment setup, including the data set and results, are described in Section 4. Finally, conclusions and future work are discussed in Section 5.

## 2. Related Work

In recent years, the proportion of elderly people suffering from age-related chronic diseases has gradually increased, and age-related chronic diseases are also among the main causes of death in the elderly. Common chronic diseases in the elderly include diabetes, hypertension, hyperlipidemia, coronary heart disease, stroke, gout, chronic kidney failure, chronic bronchitis, etc. Recent enhancements in computer technology and innovations in machine-learning techniques have contributed to the development of effective predictive and decision-making tools [8,9] that help medical experts to make effective prevention and treatment choices for various chronic diseases of aging at an early stage [10]. The prediction of chronic diseases of the elderly and their associated recommendations mainly rely on patients’ diagnostic reports, physicians’ diagnostic results, and corresponding treatment methods. Therefore, by calculating the degrees of influence of various patient characteristics on various diseases, we can infer related diseases from which patients may suffer and recommend corresponding treatment methods. The rapid development of machine-learning technology can solve this prediction problem. Data-fusion technology [11,12] has been applied to predict diabetes and make appropriate recommendations [13]. Using data fusion, the irrelevant burden of system computing power can be eliminated, and the performance of the proposed system can be improved to predict patients’ diseases and make recommendations more accurately. In order to explore a drug-recommendation method based on elderly patients with diabetes, hypertension, and cardiovascular disease [14], the collaborative filtering method was combined with a traditional machine-learning classifier to form a collaborative-filtering hybrid model, which improved the recall rate and accuracy rate, and the experimental results also showed that collaborative-filtering technology was always worse than traditional classifiers.

Assisting physicians in making medical diagnoses and reducing the likelihood of misdiagnosis is one of the main categories of the medical referral system. To reduce the risk of physicians prescribing incorrect drugs [15], based on machine-learning algorithms (neural networks, Bayesian networks) and data mining (clustering, classification) algorithms, a variety of decision-support systems or recommendation systems have been developed to assist physicians in obtaining better diagnostic results and to remind physicians of some easily ignored problems [16]. A diabetes-prediction and -diagnosis model (IFIR_PDDM) based on intelligent fuzzy inference rules was proposed to provide content recommendations for diabetes patients and predict the likelihood that current patients have diabetes [17]. A disease-diagnosis-and-treatment recommendation system (DDTRS) was proposed, which combines the DPCA algorithm and the Apriori algorithm to realize disease prediction and recommend appropriate diagnoses and treatment plans for patients and inexperienced physicians. The authors of [18] presented a simple algorithm that uses classification algorithms to help physicians predict a patient’s multivariate heart-disease risk. Currently, the healthcare-system model should have high accuracy and sensitivity, so that patients are not misdiagnosed, and help healthcare workers and patients better prevent disease and its diagnosis [19]. This emphasizes the need for skills to solve health problems with the help of AI-related technologies. Combining ML, DL, NN, the Internet of Things, and CC can help inexperienced medical staff to better predict the direction of diseases and devise corresponding treatment plans based on the current diagnoses of patients, and help medical staff identify potential dangers to patients more accurately and in a more timely manner. Timely intervention helps patients to overcome physical diseases and reduce the pain they suffer.

Healthcare is also an important application area for medical recommendations [20]. A new algorithm, fb-kNN, is proposed as a recommendation algorithm based on human-disease-rule analysis and then implemented in Healthcare 4.0 for the recommendation of diagnoses and treatments [21]. A reminder-care system was proposed to help Alzheimer’s patients live safely and independently at home. The proposed recommendation system was developed based on a contextual slot-machine approach to address the dynamics of human activity patterns to provide user-needs recommendations without user feedback. The experimental results proved the feasibility and effectiveness of the proposed reminder-care system in a real-world intelligent home application based on the Internet of Things. The authors of [22] proposed an efficient community recommendation system based on the Internet of Things to diagnose heart diseases and their types and provide suggestions related to body and diet plans to solve the difficulty of accessing doctors in remote mountainous areas [23]. A new system architecture was proposed that combines several future technologies, such as artificial intelligence (AI), machine learning (ML) and deep learning (DL), virtual reality (VR), augmented reality (AR), mixed reality (MR), and the haptic Internet (TI). The whole system involves the clinical care of patients, immediate medical testing, the diagnosis of patients’ physical conditions, and timely manual treatments, which reduces the workload of medical staff and helps them to make and accurate diagnoses. This vision of the future provides a direction for the future development of healthcare systems.

Although we have carried out many studies on medical recommendation systems in recent years, there are always some corresponding defects in these systems, such as unreasonable data collection, a lack of balance, and the fact that the importance of some features is not apparent. The non-interpretability of recommendation results is a further drawback; physicians are not involved in the process of medical recommendations, which makes patients lack trust in the system. All these aspects led us to question the recommendation results of the medical recommendation system, and we are eager to solve these problems in future research.

## 3. Proposed Method

The proposed method realizes the interpretability of the recommendation results of the traditional medical recommendation algorithm, can assist physicians and patients in making corresponding decisions in relation to disease diagnosis, explain the main factors causing the current situation, and remind patients to pay attention to related physical conditions and living habits in daily life. The proposed method is mainly aimed at solving the inexplicable problems of the traditional medical recommendation system. The architecture of the proposed interpretable model is shown in Figure 1. Data preprocessing was applied to the disease data of patients collected from Kaggle, deleting partial data, incomplete, digitized text data. Next, the processed data were respectively trained by naive Bayes algorithm [24], logistic-regression algorithm [25], decision-tree algorithm [26], random-forest algorithm [27], gradient-enhanced-tree algorithm, and multi-layer perceptron [28]. Finally, the results after training were analyzed by LIME algorithm [29,30] to explain the reasons for recommending the results. The details of each method are described in the subsequent sections of this chapter.

### 3.1. Data Preprocessing

Data preprocessing is a critical step in the data-analysis process, as the quality and accuracy of the final recommendation results depend on the information contained in the data. To ensure reliable and accurate analyses, several preprocessing techniques are applied. Firstly, duplicate data, which refers to identical or highly similar records, is identified and removed from the dataset. This eliminates bias and prevents inflated accuracy. Secondly, missing data are addressed by imputing missing values or excluding records with excessive missing data, thus maintaining the dataset’s integrity. Unrealistic data, such as outliers, are detected and treated to mitigate their influence on subsequent analyses. Additionally, text data are digitized using one-hot encoding or word-embedding techniques, allowing numerical representation. Although the preprocessed data are ready for model training, to prevent overfitting, the zero-mean normalization method is employed to normalize the data, ensuring comparability between different features. This approach, mentioned in [31], aids in standardizing the data and improving the performance of subsequent modeling tasks. These data-preprocessing steps enhance the analysis results’ reliability, consistency, and quality, enabling robust and accurate recommendations.

### 3.2. Method

The normalized data set is mapped to the training model, and the algorithm learns the relationship between each data feature and label in the training set to achieve disease prediction. Experts labeled the data set used in this study, and the target variables have two classes. A 0 means no diabetes or heart disease, and 1 implies diabetes or heart disease. This process helps discover a patient’s likelihood of developing a condition based on their selected key characteristics.

This study uses six different classical algorithms to train the model: naive Bayes algorithm, logistic-regression algorithm, decision-tree algorithm, random-forest algorithm, gradient-enhanced-tree algorithm, and multi-layer perceptron. Next, the test data set is applied to the trained model to obtain the test performance of the model. By running the model repeatedly and changing the data in the test set and training set, the accuracy of the model is determined. Precision indexes are used to evaluate the prediction results. These classical algorithms are briefly discussed in the following sections.

#### 3.2.1. Naive Bayes (NB) Algorithm

Bayesian algorithm is a method to classify sample data sets based on Bayesian principle, using the relevant knowledge of probability statistics. Because the Bayesian algorithm has a solid mathematical and theoretical foundation, the error rate of the algorithm is relatively low. The Bayes formula is as follows:(1)$$P(X|Y)=\frac{P\left(XY\right)}{P\left(Y\right)}=\frac{P(Y|X)P\left(X\right)}{P\left(Y\right)},$$

In other words, the posterior probability is expressed by prior probability and conditional probability to calculate the posterior probability. However, because of its complex dependent relation, the difficulty in using Bayes algorithm in real operations increases sharply, especially in multi-feature conditions, so naive Bayes algorithm [32] is applied. Naive Bayes algorithm is a classification method based on Bayes’ theorem and the assumption of independence of characteristic conditions; that is, it is assumed that all attributes associated with a given target value are independent of each other [33,34]. Thus, the conditional probability function is converted into the product form of characteristic conditions of each dimension. The formula is as follows:(2)$$P(x|y_{k})=P(x_{1},x_{2},\cdots ,x_{d}|y_{k})=\prod_{i}^{d}P(x_{i}|y_{k}),$$

Therefore, the naive Bayes formula can be derived as follows:(3)$$P(y_{i}|x_{1},x_{2},\cdots ,x_{d})=\frac{P\left(y_{i}\right)\prod_{j=1}^{d}P(x_{j}|y_{i})}{\prod_{j=1}^{d}P\left(x_{j}\right)},$$
where $y_{i}$ represents the category and $x_{j}$ represents the item’s fourth characteristic. In other words, no characteristic variable has an absolute proportion in the decision result, and no characteristic variable has a small proportion in the decision result. Although this simplified method reduces the accuracy and reliability of Bayes classification algorithm to a certain extent, it greatly simplifies the complexity of Bayes method in practical application scenarios, so the naive Bayes algorithm can really be applied in complex scenarios, and shows powerful computing performance.

#### 3.2.2. Logistic Regression (LR) Algorithm

Logistic regression (regression) [35,36] is a widely used binary algorithm for machine learning. It is the simplest of all machine-learning algorithms, featuring fast prediction speed, and easy learning and understanding. Logistic regression is similar to linear regression in that linear regression requires the dependent variable to be continuous, while logistic regression requires the dependent variable to be categorical. Logistic-regression algorithm mainly includes the following steps.

First, the corresponding logistic-regression multivariate equation is constructed as the hypothesis function according to the number of sample features in the data set. The input data features of samples are expressed as the independent-variable form of the function to obtain the importance of each feature to the classification results and the corresponding function bias. The specific formula is as follows:(4)$$z=w_{0}x_{0}+w_{1}x_{1}+\cdots +w_{n}x_{n}+b,$$

The second step is to input the data features obtained in the previous step into the logistic-regression model to calculate the classification probability. Sigmoid logistic-regression model is commonly used in classical logistic-regression models. The regression-model formula of specific series is as follows:(5)$$S\left(z\right)=\frac{1}{1+e^{-z}},$$

The sigmoid function is an *S*-shaped curve that can map any real number to a value between 0 and 1.

The third step is to update the linear-regression parameters using cross entropy, other loss functions, and gradient-descent algorithm. The cross-entropy loss function is as follows:(6)$$L\left(w,b\right)=\prod_{i=1}^{m}{\left(S\left(z\right)\right)}^{y_{i}}{\left(1-S\left(z\right)\right)}^{1-y_{i}},$$
where $y_{i}$ represents the classification of results (Examples: $y_{i}=1$ represents a positive example, $y_{i}=0$ represents a negative example).

According to the above steps, appropriate model parameters can be trained, and then the trained parameters can be used to achieve the logistic-regression classification of data features.

#### 3.2.3. Decision Tree (DT) Algorithm

Decision tree algorithm [37] is a method to approximate the value of discrete function. It is a typical machine-learning algorithm that first processes data, uses inductive algorithms to generate readable rules and decision trees, and then uses decision to analyze new data. A decision tree is a process of classifying data by a set of rules. The decision-tree algorithm constructs a decision tree to find the implied classification rules in the data. Construction of a decision tree with high precision and on a small scale is the core aim of decision-tree algorithms. There are various methods to construct decision trees, among which ID3 [38] is a classic method, the core of which is to use the concept of “information entropy.” The information-entropy formula is as follows:(7)$$Entropy\left(t\right)=-\sum_{i=0}^{c-1}P(i|t)log_{2}P(i|t),$$
where P(i|t) represents the probability that node *t* is classification *i*, *log*_2 is logarithm base two. Entropy(t) reflects the uncertainty of this information. The greater the uncertainty, the greater the amount of information it contains, and the higher the entropy of information. The ID3 algorithm calculates information gain, which refers to the increase in purity and decrease in information entropy caused by partition. It is calculated by the information entropy of the parent node minus the information entropy of all the children. The formula is as follows:(8)$$\mathrm{Gain}\left(D,a\right)=\mathrm{Entropy}\left(D\right)-\sum_{i=1}^{k}\frac{\left|D_{i}\right|}{\left|D\right|}\mathrm{Entropy}\left(D_{i}\right),$$
where D is the parent, $D_{i}$ is the child; a in $\mathrm{Gain}\;\left(D,a\right)$ is selected as the attribute of D node.

The process of decision-tree construction can be divided into two steps. The first step is the generation of decision tree: the process of generating a decision tree through training-sample data. In general, the sample data set must be real and reliable and contain rich data features, and it is necessary to attempt to make the data features balanced and comprehensive. The second step is pruning the decision tree: pruning the decision tree is a method to prevent data overfitting in decision trees. It is a process of checking and modifying the decision tree generated in the previous stage. It mainly uses the data in the new test set to check the decision rules generated during the generation of the decision tree. It cuts off some branches that affect the accuracy of the balance.

#### 3.2.4. Random Forest (RF) Algorithm

Random forest is an integrated learning algorithm composed of multiple decision trees and a supervised learning algorithm. Random forest is used as a classification and prediction model in many fields. None of the decision trees in a random forest are connected to each other. The entire random forest is constructed by importing the test set in the data set into the random forest model for the generation of each decision tree [39]. Next, the test set in the data set is imported into the random forest created in the previous stage, and each decision tree in the forest is predicted in turn. Thus, the prediction result of the user in the current decision tree is obtained. After stablishing the most frequently predicted result of the current user from all decision trees, the result is taken as the final prediction result of the random forest of the current user.

Although random forest appears similar to building blocks from this point of view, as the classification tree grows, random forest only considers binary split points on random subsets of contributing variables, so the accuracy is significantly improved over that of a single classifier, especially in relatively complex models. The size of randomly selected subset variables is predefined.

#### 3.2.5. Gradient Boosting (GB) Decision Tree

Gradient-enhancement decision tree [40] is a machine-learning algorithm for classification and regression problems. It is one of the ensemble-learning algorithms that combine multiple weak models to create a strong model that can make accurate predictions. This algorithm is a very effective technique, which has been applied to various machine-learning problems and obtained quite good results.

The gradient-enhancement algorithm is implemented by continuous training of some weak models. It iteratively trains a series of decision-tree models, each attempting to correct the errors of the previous model. In each iteration, the model adjusts the weight of the sample based on the difference (gradient) between the current model’s predictions and the true label, so that the next model pays more attention to the samples misclassified by the previous model, so as to ensure that the objective function gradually improves the performance of the overall algorithm with the progress of the calculation depth. This optimization algorithm can reduce the calculation errors of the model by adjusting the weight vector of the features in the model. The specific algorithm flow is shown in Algorithm 1:
**Algorithm 1:** Gradient Boosting$\mathrm{initialization}\;F_{0}\left(x\right)={\arg\;\min}_{h\subset H}\mathrm{Loss}\left(y_{i},h\left(x_{i}\right)\right)$**For** k=1:  $\mathrm{Calculate}\;\mathrm{the}\;\mathrm{negative}\;\mathrm{gradient}\;\mathrm{of}\;\mathrm{the}\;\mathrm{cumulative}\;\mathrm{model}\;\mathrm{loss}\;\mathrm{function}\;g_{k}=-\frac{\delta \mathrm{Loss}\left(y,F_{k-1}\left(x\right)\right)}{{\delta F}_{k-1}\left(x\right)}$  $\mathrm{The}\;\mathrm{fit}\;\mathrm{weak}\;\mathrm{learner}\;\mathrm{makes}\;\sum_{i=1}^{N}{\left(g_{k}^{i}-h\left(x_{i}\right)\right)}^{2}$  $\mathrm{Updating}\;\mathrm{cumulative}\;\mathrm{model}\;F_{k}=F_{k-1}+\alpha h\left(x\right)$ where α is the learning rateUp to an iteration-termination condition, $\mathrm{return}\;F\left(x\right)=F_{k}\left(x\right)$

The basic working principle of gradient-lifting decision tree is as follows. Firstly, a simple model (such as a single decision tree) is used as the initial model. This initial model is then used to predict the results of the training sample and calculate the residual (error) between the predicted value and the true value. Next, based on the residual, a new model (usually a decision tree) is trained to expect the residual. By adding the prediction result of the new model to the prediction result of the previous model, an updated prediction result is obtained. New models are trained iteratively, each time trying to correct the residuals of the previous round of models until a specified number of iterations is reached or a certain stop condition is met. Finally, all the trained models are combined to obtain the final model. When forecasting, the final forecast result is obtained by adding the predicted results of each model.

The key element in gradient-lifting decision tree lies in model training and updating of prediction results in each iteration. Each new model tries to correct the residuals on the basis of the previous model, constantly adjusting the weight of the model through the method of gradient descent to minimize the residuals. Gradient-enhanced decision trees have good performance and flexibility in ensemble learning, and can deal with complex classification and regression problems. They can optimize a model’s performance and robustness [41] by adjusting parameters such as learning rate, number of trees, depth of trees, etc. However, it is easy to overfit training data with GPD, so it is necessary to adjust parameters and use cross-validation methods to control the complexity and generalization ability of the model.

#### 3.2.6. Multilayer-Perceptron Algorithm (MLP)

Multilayer perceptron [42] is a famous feedforward artificial neural network, which has achieved great success in data-classification applications. It consists of a simple node called a perceptron, capable of producing a single output from multiple inputs by assigning weights to inputs and creating linear combinations of weighted inputs. The output is then computed using a nonlinear kernel function.

The MLPS are trained using a backpropagation technique consisting of two stages. The first stage is forward transmission, in which a given input datum is given a classified output. The second stage is the reverse transfer, in which the partial derivative of the kernel corresponding to the changing parameter is calculated and then diffused back into the network. The weight of the network can then be adjusted using any gradient-enhancement algorithm, and the process can be repeated until the weight of the network converges.

### 3.3. Interpretable Model Based on LIME

The key part of the hybrid model proposed in this study is the explainable model, and the explainable method is independent of the explainable model. Its aims are to realize the interpretation of the recommendation results of a variety of classical classification-recommendation algorithms, to inform patients as to the reasons for the recommendation, so that patients fully accept the recommendation, and to make the recommendation system truly achieve the purpose of disease prediction and prevention.

In the previous recommendation system, the authors often used the recall rate and accuracy rate of the test set to measure the quality of the model, but the vague recommendation results prevented patients from trusting the recommendations of the system. The system needs to explain the reason for its recommendation results. However, not all models, such as convolutional neural networks, can explain the recommendation results. The most commonly used interpretation methods are the use of the model coefficient of linear regression or the feature importance of the decision tree to explain the recommendation results. However, these methods are affected by the original algorithm model, and when the algorithm model is relatively complex, they cannot be used to effectively explain the recommendation results. Therefore, in order to realize the explainability of complex models, we introduce an explainability method that models in this study do not constrain.

The explainable method used in this study is the LIME explainable model. The model is a locally explainable method independent of the model itself. A trained local-agent model is used to interpret a single sample. It is assumed that for a black-box model that needs to be explained, a relevant instance sample is taken, a new sample point is generated by perturbation near it, and the predicted value of the black-box model is obtained. The new data set is used to train the interpretable model (such as linear regression and decision tree), and a good local approximation to the black-box model is obtained.

The idea behind LIME is very simple. Its main purpose is to explain complex models through simple models, and LIME simply explains each sample itself. It obtains a new sample data set through data transformation on a certain text. It then performs simple model training on this new sample data set, and it is hoped that the prediction results of this simple model can approximate the prediction results of the more complex model on the original data set. In order to be interpretable and unconstrained by the model, LIME does not operate deep in the model.

Therefore, LIME’s steps are as follows: train the whole model (which itself is not explainable); select the variables to be explained; and N perturbations can be made to the data in the data set to generate local samples. The new local sample is weighted. The weight is the distance between these data points and the data to be interpreted. According to the new data, a simple model is fitted. Next, a simple model is used to explain the complex model at a certain sample point.

## 4. Experimental Results and Discussion

The data set used in this experiment, the details of the experimental setup of the model, and the experimental results obtained are explained in detail in the following sections.

### 4.1. Data Set

In this study, two data sets were used, namely the heart-disease data set and the diabetes data set, through which the prediction of corresponding chronic diseases of old age was planned. The data set was from the Kaggle website. The aim was to calculate the characteristics of the data set, which included a range of factors, such as the patients’ daily habits, test reports, and environment, to predict whether the patients would have a disease (heart disease or diabetes). The heart disease-dataset contained the characteristics of a total of 46,783 patients with and without a diagnosis of heart disease. The heart-disease dataset was a processed dataset, containing 253,680 survey responses to CDC BRFS 2015. The target variable had two categories, no heart disease and heart disease, and the dataset contained 21 characteristics. The diabetes data set contained the medical records of 768 patients, and the target variables included whether or not the patients had diabetes, with 0 representing no diabetes and 1 representing diabetes.

With regards to the heart-disease data set, we adopted a stratified sampling technique to collect the data set samples, mainly based on gender, age, blood pressure, heart history, the amount of exercise, and the smoking habits of the patients. These features were selected because they are known to play an important role in the diagnosis of heart-disease types based on clinical evidence. The different nature of these data helps to provide a common and adaptable system for medical specialists. The data set included 56% men and 44% women, about 58% of whom had hypertension and about 42% of whom did not. Table 1 provides a detailed description of some of the features in the data set.

The diabetes dataset was created by the National Institute of Diabetes and Digestive and Kidney Diseases of the United States. The dataset was collected to predict whether a patient would have diabetes based on certain diagnostic measures contained in the dataset, but the sample size of the dataset was small. The dataset had an age range of between 25–65, and all the patients were female, with the same Pima Indian ancestry.

This model not only outputs the likelihood of a patient having a disease, but also illustrates the contribution rates of various features when the system makes a recommendation, reminding the patient to make changes in the corresponding areas.

### 4.2. Evaluation Parameters

According to the research plan, the data set was divided into two parts, the training set and test set, and the model as trained and tested, respectively. In the heart-disease data set, the training set accounted for 80 percent of the total, comprising 37,426 patients, and the test set accounted for 20 percent of the total, comprising 9357 patients. The diabetes data sets were also segmented in this way.

Six classical classification algorithms, such as random forest, multi-layer perceptron, and logistic regression, were used for recommendations related to medical diseases. On this basis, the LIME model was used to reconstruct the data set for predicting the characteristics of the patients, so as to realize an interpretable analysis of the recommendation results. Five properties were used for these evaluations: accuracy, computation time, LIME computation time, the LIME interpretation model, and the LIME local predictions versus the actual predictions.

The aims of this study are to make it easier for patients to understand the reasons why the system recommends particular results by explaining the results of medical-disease recommendations and to make it easier for patients to accept the recommended results and take corresponding measures to solve their problems. The criteria used to measure the performance of conventional recommendation systems include accuracy and coverage, among others. In this study, we planned to realize make the recommendation results interpretable without affecting or while only slightly affecting the criteria, such as accuracy.

### 4.3. Results

This section introduces the performance of a LIME-based interpretable recommendation system in terms of prediction accuracy, computational cost, white-box-model interpretability, and LIME-based interpretability. The interpretable computational cost of the black-box model and the performance comparison between the interpretation results of the black-box model and the white-box model are calculated, and the model-interpretation performance of LIME is explained.

#### 4.3.1. Accuracy

Accuracy is an important performance indicator in recommendation systems, and it is unacceptable to reduce the accuracy in all recommendation systems significantly. Accuracy is an important indicator for the adoption of system recommendations. Relatively complex models can be improved by relatively high accuracy; we conducted accuracy-analysis experiments on six models, including decision tree and linear regression, and the results are shown in Figure 2. It can be seen from the results that the more complex the model, the higher its test accuracy. It can also be seen from the difference between its training accuracy and test accuracy that the relatively complex model is also relatively stable. This was consistent with our speculations.

Similar to the accuracy of the above experiments, we speculated that the more complex the model, the higher the time cost would be. The experimental results are shown in Figure 3. According to the figure, the more complex the model, the longer the training time and the higher the training cost.

#### 4.3.2. White-Box Interpretable Model

The medical-recommendation system is exceptional, and fundamentally different from the recommendation systems of movies, books, searches, etc. Patients do not use the recommendation system when their attitudes are inappropriate. For patients to fully trust the recommendation system, it must be sufficiently persuasive. The system must give a convincing reason for the recommendation. If this is the case, patients accept the system’s recommendation results, and the medical-recommendation results provide value. Due to the characteristics of the models themselves, some traditional recommendation algorithms can make use of their own characteristic information to make recommendations interpretable; the most common approaches is the model coefficient of the logistic regression model and the linear regression model, whose coefficient itself can explain the degrees of the contributions of different features to the results. The results are shown in Figure 4, where the roles of various characteristics in this patient-recommendation outcome are displayed. (a) The results showed that the current patient did not have heart disease, mainly due to the excellent performance of Genhlth, BMI, age, and HighBP. However, the results should also alert patients to HvyAlcoholConsump. (b) The results showed that the current patient did not have diabetes, mainly because the characteristics of glucose, BMI, age, and diabetes pedigree function were good, but the patient should pay attention to their blood pressure and insulin.

Another way to explain the medical-recommendation system is the feature importance of the decision tree. The explainable range of this method is called the regression model, and its explainable range is relatively wide. Most models that use decision trees are interpretable. The experimental results are shown in Figure 5, where (a) the interpretable results of the heart-disease dataset and (b) the interpretable results of the diabetes dataset are shown. It can be seen that the decision-tree method can analyze the feature importance of the decision tree, random forest, and gradient-enhancement models and provide a feature-importance analysis of the recommendation results in the data set, so as to explain the degree of the contribution of each feature to the recommendation result, which can be used as explanations for recommendation results and provide reasons for patients to accept these results.

From the above, it can be seen that explainability exists, and that it is sufficient for patients, which makes the recommendations acceptable. However, it can also be seen that this approach has some limitations and cannot explain other models, limiting the explainability scope. Therefore, LIME is introduced into medical recommendations to provide explanations for relatively complex models and for models that do not offer explanations.

#### 4.3.3. Interpretation Based on LIME

Although the models in the current medical-recommendation system are becoming increasingly complex and their accuracy is gradually increasing, their non-interpretability is also becoming prominent. Therefore, LIME is used in this study to realize the interpretability of relatively complex models. The LIME model mainly interprets a single model by training a local proxy model, takes concerned instance samples, generates new sample points through perturbation near them, and obtains the predicted values of relatively complex models. New data sets are used to train interpretable models (such as linear regression and decision trees) to obtain good local approximations to these models. The interpretable recommendations are then implemented using a similar model interpretation instead of complex model interpretation.

In this study, we implemented interpretability recommendations for a relatively simple classification model (naive Bayes) and interpretability interpretations for a relatively complex multilayer-perceptron model through LIME based on data sets for heart disease and diabetes. The experimental results are shown in Figure 6. We found that regardless of whether it is a simple classification model or a relatively complex model, and of whether it is self-explained or explainable by external models only, the LIME model can offer the interpretation of its recommendation results. Although the essential features of the model that different classification models can explain are varied, from the above, it can be seen that the contribution of these features to the results is positive or negative and is about the same as in the alternatives. This can help us understand which traits are essential and should be retained, and which traits make a low contribution to current disease and can be removed.

According to the information presented in the figure above, this paper summarizes the important features and corresponding values of each algorithm in the data set of diabetes and heart disease, so that the importance of each feature can be more clearly understood. The specific data are shown in Table 2.

Although we know from the above that the LIME model can implement the recommended interpretation of the unexplainable model itself, we are focused not only the interpretation itself, but also on the running cost of the interpretation. Not only must a good model achieve its original practical purpose, but its operating cost must also be within an acceptable range. Therefore, we also collected the running time of the individual test data of the model in the interpretability experiment, and fully calculated the interpretability result of the current model by timing it before creating the interpreter and ending with the recommended interpretability. Thus, the running time of the entire LIME model’s interpretation was obtained, representing the running cost of the model. As shown in Table 3 of the experimental results, the time required for naive Bayes classifier in the heart-disease data set was almost the same as that required for the multilayer perceptron. In contrast, logistic regression requires the highest running costs, and the time required for random-forest operations is even less than that required for decision trees. In the diabetes data set, it can be seen that the gradient-enhancement algorithm and random forest take significant time, and that the MLP takes a relatively short time. It can be concluded from the above table that for complex models, the operation time required by LIME for local interpretation does not increase significantly, the time required by different models is little different, and the use time of single-test data are relatively small. Therefore, the LIME model has relatively low operating costs and occupies fewer operating resources, making it suitable for recommendation and interpretation in relatively complex models.

While LIME can explain the characteristic importance of these results, we are still unable to draw conclusions. We need to check whether the local model really approximates the original model closely, and to judge whether the LIME local model can replace the model of the recommendation system for recommendation by comparing the difference between the LIME local prediction and the actual prediction. The experimental results are shown in Figure 7a, where green represents the LIME local predictions and brown represents the actual predictions. It can be seen that the actual predicted values of the naive Bayes model, logistic regression, multilayer perceptron, and gradient-enhancement tree were little different with the local prediction effect of LIME, and the trend was consistent with the actual predicted value. Only the LIME prediction for decision tree was significantly different from the actual predicted value. As shown in Figure 7b, the LIME local predicted values of the logistic regression, random forest, gradient enhancement, and multilayer perceptron were little different from the actual predicted values, so they can be used as their approximate models. Therefore, from these experimental results, it can be inferred that the local model can be approximated as an alternative to the original model in LIME interpretation to ensure the interpretability of medical recommendations.

In order to more clearly show the advantages of the current algorithm model for the data sets of diabetes and heart disease, we compared the algorithm with other algorithms. We found that few medical-recommendation algorithms about heart disease are used at present and that, in the existing heart-disease recommendation system, there were almost no corresponding explanations for the recommended results. Similarly, although more research has been conducted on diabetes than on heart disease, the data sets on diabetes were rarely able to explain the results related to the disease. The details are shown in Table 4, below.

### 4.4. Discussion

This paper proposes a deep-learning approach combined with local interpretable model–agnostic explanations (LIME) to address the lack of explanations in health–recommendation systems, particularly for chronic diseases in older adults, such as heart disease and diabetes. The results obtained from applying this approach to the heart-disease and diabetes datasets are discussed, highlighting the similarities between the actual model predictions and the local model recommendations generated by LIME.

For the heart-disease dataset, both the multilayer perceptron and the gradient-boosting algorithm demonstrated consistent predictions with the local-model recommendations of LIME. This indicates that the local model generated by LIME can serve as an approximate prediction for heart disease. Furthermore, the feature-importance analysis revealed that the features CholCheck, GenHith, and HighBP were identified as the most important factors for heart-disease prediction. These insights provide valuable information to medical practitioners for understanding the key variables contributing to heart disease and making informed treatment decisions.

In the case of the diabetes dataset, both the multilayer perceptron and the logistic-regression algorithm showed similar predictions to the local-model recommendations of LIME. This suggests that LIME’s local model can provide a reasonable approximation for recommending treatment options for diabetes. The feature-importance analysis identified glucose, BMI, and age as the most influential features in predicting diabetes. Understanding the importance of these variables allows medical professionals to focus on the crucial factors affecting diabetes and to personalize patient-management strategies accordingly. By using LIME, this paper successfully determines the importance of each feature in influencing the calculated model’s results. The contribution coefficients of these features are presented, offering insights into the relative impact of different patient characteristics on the final recommendations. This approach enhances patient trust by providing understandable and transparent explanations for the recommendations made by the system. It bridges the gap between the black-box nature of deep-learning models and the need for interpretability in medical decision making.

The proposed system has significant implications for medical-recommendation systems and promotes informed decision making in healthcare. By elucidating the underlying reasons behind recommendations, medical practitioners gain valuable insights into the decision-making process and can engage in more effective communication with patients. This approach fosters a collaborative and informed healthcare environment, empowering patients and improving overall treatment outcomes. However, it is important to note that this paper lacks detailed descriptions of the modifications made to the LIME method and the specific methodology employed. Further clarification and details regarding the experimental setup, evaluation metrics, and comparison with existing approaches would strengthen the study. The model studied aims to use LIME’s explanatory techniques to provide explainable reasons for recommendations for more complex recommendation models. The results show that the proposed method can provide interpretability for many models that are not interpretable by themselves, with good accuracy and computation, and that it has high scalability. This study is relatively reliable and accurate, because the data set it uses was collected by professional institutions, and the experimental results are trustworthy. In addition, most recommendation systems in the medical and health fields only contain recommendation systems, and few systems can explain the reasons for these recommendations.

## 5. Conclusions

In order to improve the unexplainable recommendation results in the field of medical recommendations, this paper proposes an explainable medical-recommendation system for age-related chronic diseases based on LIME. By combining the LIME explainable model with traditional classification algorithms and applying it in the medical field, the reasons for recommending results are explained on the basis of realizing predictions related to chronic diseases in the elderly. In this study, the method was applied to data sets on heart disease and diabetes in elderly patients with chronic diseases. Firstly, the data sets were preprocessed and normalized to ensure that the data in the data set were true, effective, and available. Next, the sorted data sets were input into six commonly used classification-algorithm models, including decision tree, random forest, the linear-regression model, multilayer perceptron, the gradient-enhancement-tree algorithm, and naive Bayes, to propose recommendations for age-related chronic diseases. Finally, the processed data set and recommendation results were trained through the LIME model to obtain the reasons for the recommendation results and provide interpretations of these results. The experimental results show that the proposed method can be used to interpret the recommended results in both non-interpretable and interpretable classification models. In addition, in LIME, the sample size affects the expected value and average forecast. Therefore, the number of samples can be reasonably controlled to achieve a more accurate interpretation effect.

In the future, more complex classification models will be added to predict the outcomes of chronic diseases in the elderly, since relatively complex models can have better prediction accuracy. This idea will direct future research.

## Figures and Tables

**Figure 1 diagnostics-13-02681-f001:**
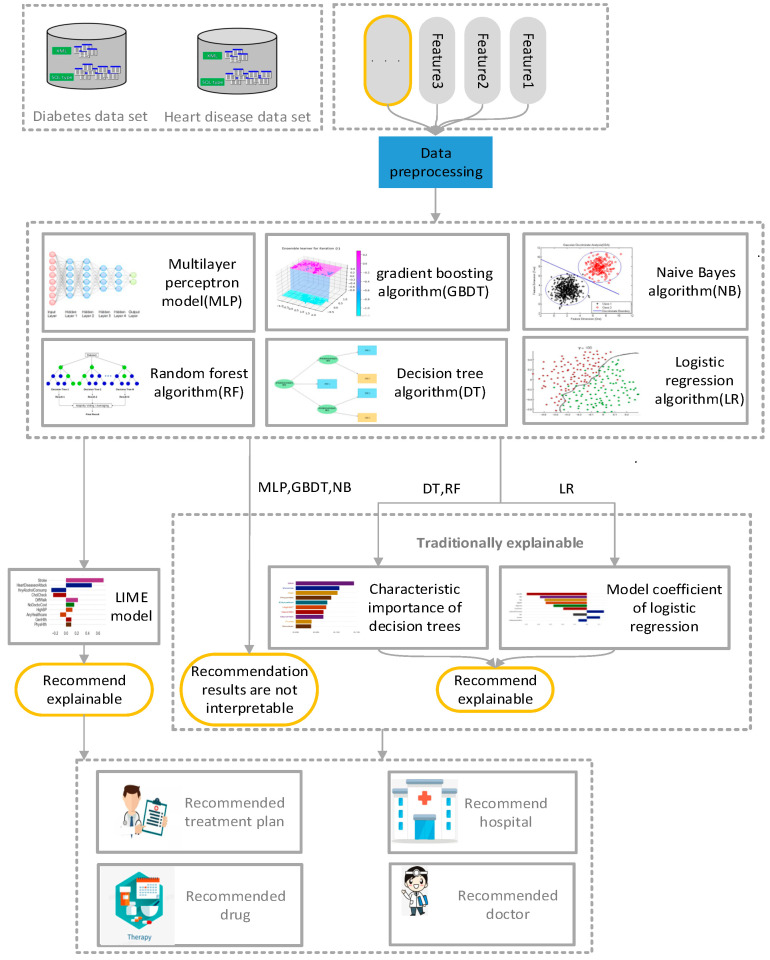
Explanation of the architecture of the model.

**Figure 2 diagnostics-13-02681-f002:**
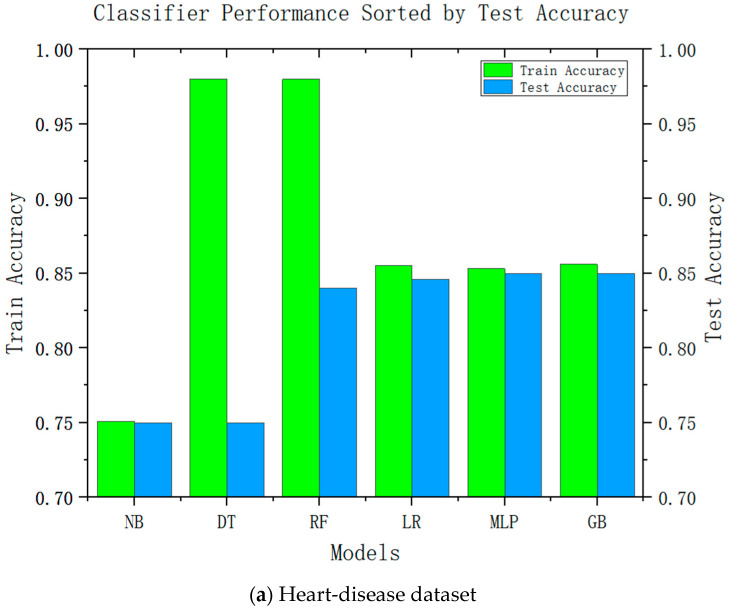
Test accuracies of six classification algorithms.

**Figure 3 diagnostics-13-02681-f003:**
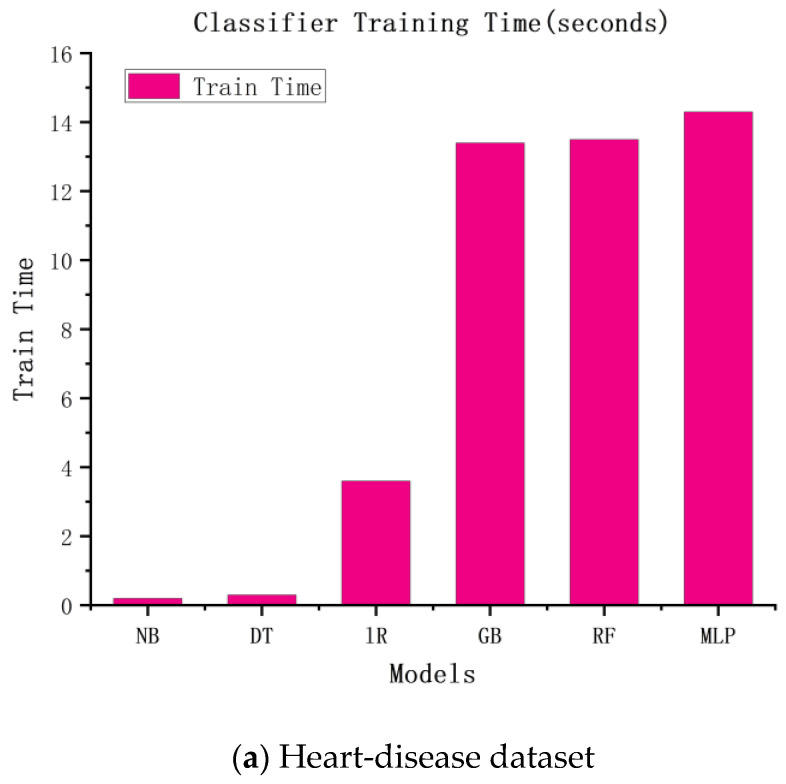
The training times of six classification algorithms.

**Figure 4 diagnostics-13-02681-f004:**
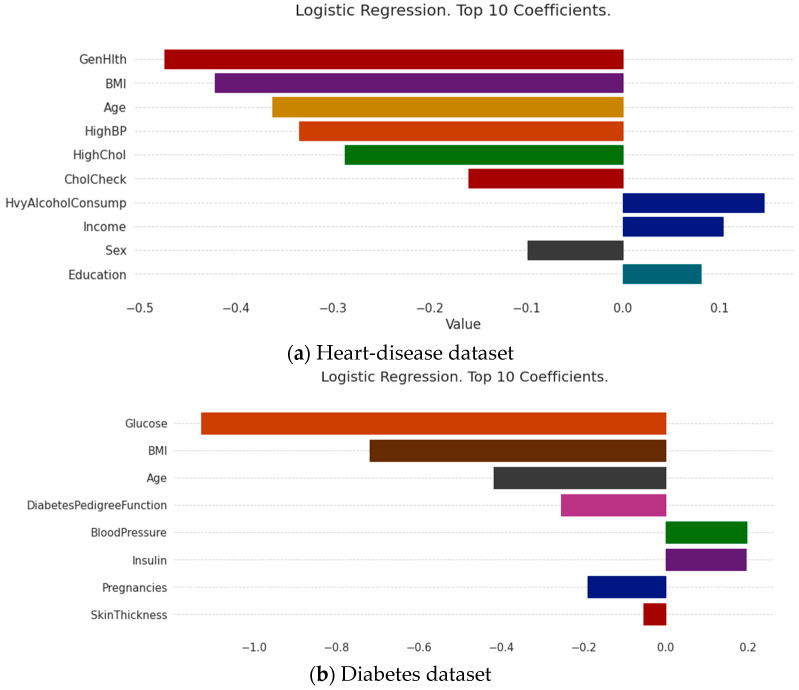
Model coefficient interpretation based on logistic regression.

**Figure 5 diagnostics-13-02681-f005:**
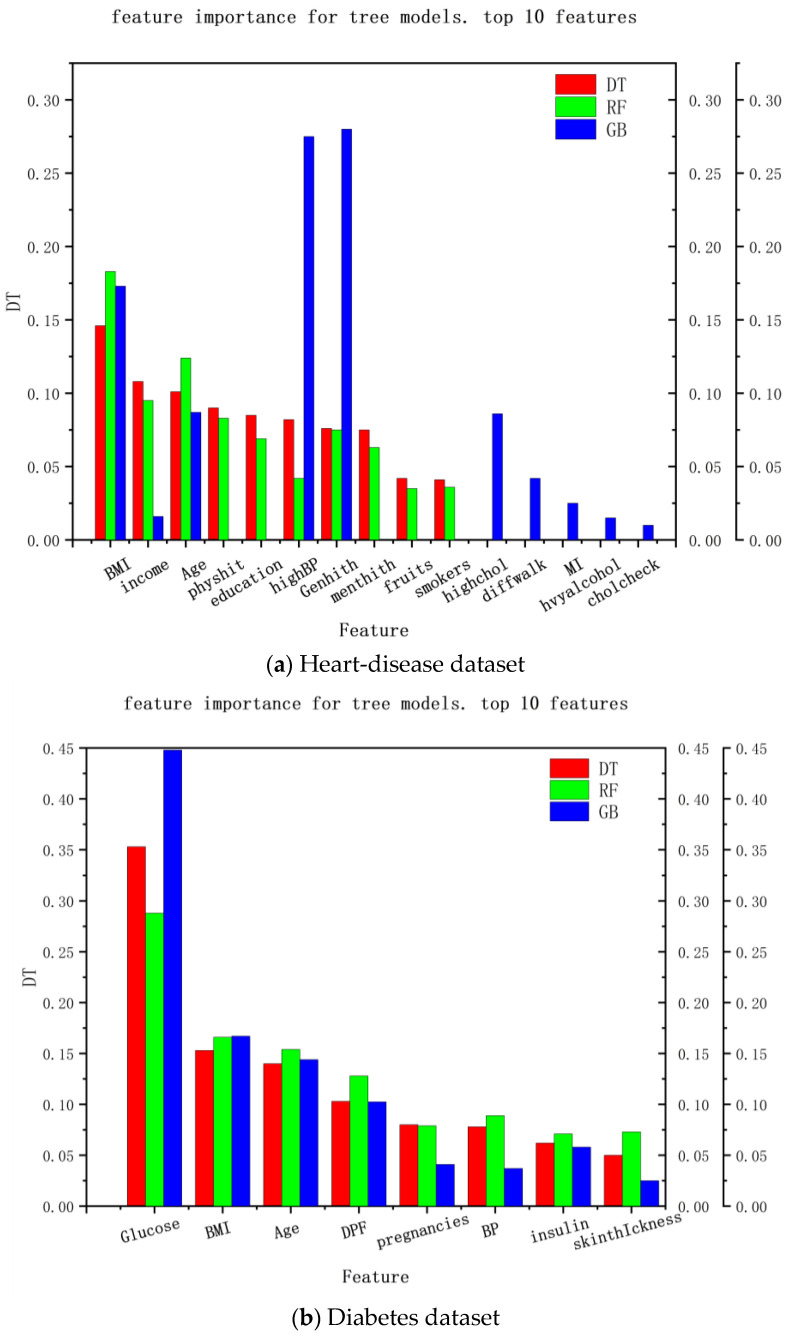
Interpretation of features’ importance based on the decision tree.

**Figure 6 diagnostics-13-02681-f006:**
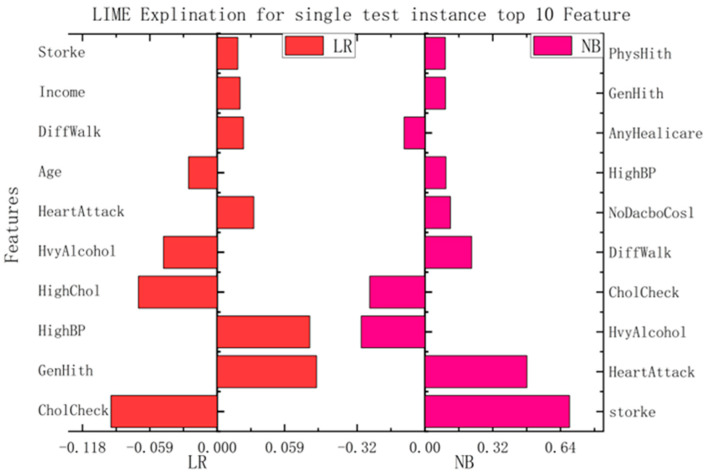
Interpretation model based on LIME.

**Figure 7 diagnostics-13-02681-f007:**
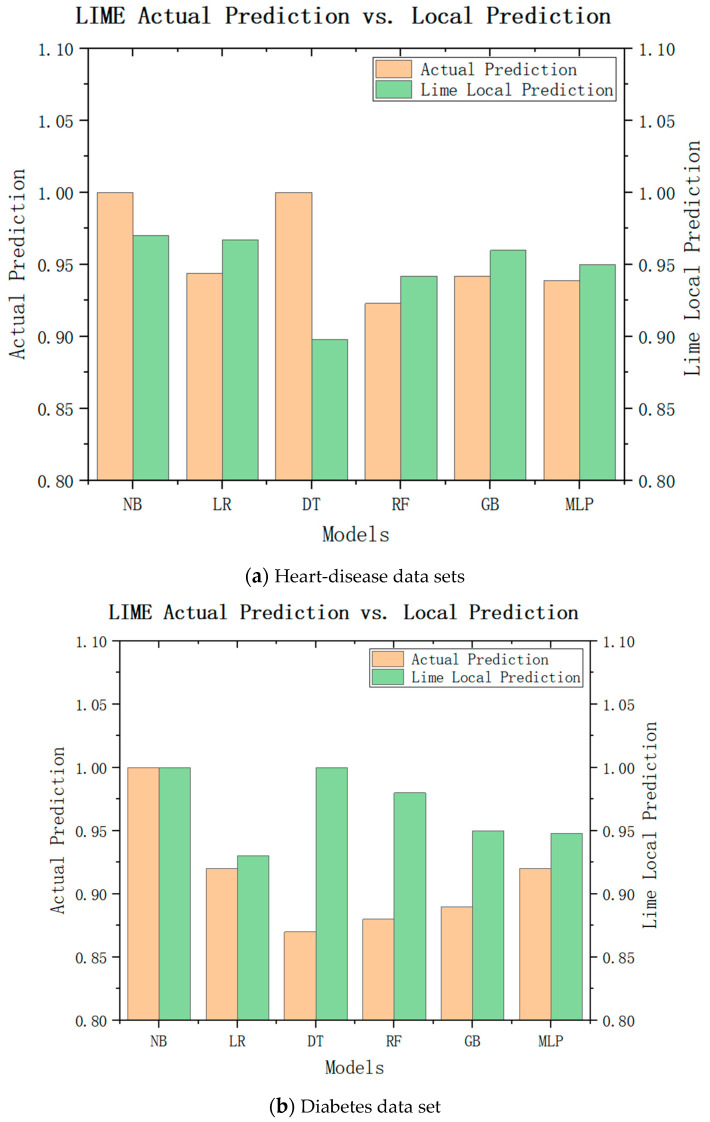
LIME local forecast and actual forecast.

**Table 1 diagnostics-13-02681-t001:** Performance-evaluation results.

Feature	Expression
Heart Attack	0 = diabetes-free 1 = diabetes
High BP	0 = nonhypertensive 1 = hypertension
High Chol.	0 = no high cholesterol 1 = high cholesterol
Chol. Check	0 = cholesterol was not tested in previous 5 years 1 = cholesterol tested within previous 5 years
Smoker	Have you smoked at least 100 cigarettes in your life? 0 = no 1 = yes
Stroke	Have you had a stroke? 0 = no 1 = yes
Diabetes	coronary heart disease (CHD) or myocardial infarction (MI) 0 = no 1 = yes
Phys. Activity	physical activity in the past 30 days—excluding work 0 = No 1 = Yes
Fruits	eat fruit one or more times a day 0 = No 1 = Yes

**Table 2 diagnostics-13-02681-t002:** Important eigenvalues based on LIME interpretation.

(**a**) Heart-disease dataset						
	**MLP Classifier**	**Gradient Boosting**	**Random Forest**	**Decision Tree**	**Naive Bayes**	**Logistic Regression**
GenHith	0.118	0.068	0.074	0.071	0.097	0.087
HighBP	0.079	0.065	0.083	0.071	0.099	0.081
HighChol	−0.071	−0.052	−0.069	−0.070	0.002	−0.069
Heart disease or attack	0.065	0.041	0.075	0.056	0.481	−0.032
Chol Check	−0.119	−0.082	−0.047	0.038	−0.26	−0.093
(**b**) Diabetes dataset						
	**MLP Classifier**	**Gradient Boosting**	**Random Forest**	**Decision Tree**	**Naive Bayes**	**Logistic Regression**
Glucose	0.278	0.348	0.224	0.348	0.229	0.284
BMI	0.208	0.194	0.143	0.127	0.179	0.211
Age	0.117	0.154	0.131	0.112	0.116	0.095
Diabetes pedigree function	0.061	0.015	0.014	0.062	0.081	0.014
Insulin	−0.023	0.002	0.013	0.006	0.072	−0.022
Pregnancies	0.047	0.028	0.015	−0.028	0.089	0.042

**Table 3 diagnostics-13-02681-t003:** Runtime (seconds) for single-test-data-instance LIME explanation.

Method	Heart Disease	Diabetes
MLP classifier	2.89 s	0.312 s
Gradient boosting	3.08 s	0.42 s
Random forest	3.11 s	0.433 s
Decision tree	3.57 s	0.368 s
Naive Bayes	3.64 s	0.366 s
Logistic regression	4.29 s	0.371 s

**Table 4 diagnostics-13-02681-t004:** Comparison with current studies.

Dataset	Study	Method	Feature	LIME	Heart Disease	Diabetes
Public dataset (PIMA)	Nagaraj, Palanigurupackiam [43]	IFIR, K-NN	Important signs, age, weight, and pricing data, etc.	×	×	√
Public dataset	Shadi Alian; Juan Li, et al. [44]	Logic-based rules	Saturated fat, dietary cholesterol, trans-fat, etc.	×	×	√
Hospital dataset	Anam Mustaqeem, et al. [45]	Statistical analysis	Diastolic blood, age, BMI etc.	×	√	×
Hospital dataset	Mengxing Huang et al. [46]	CML-KNN, a k-nearest neighbors	CT, MRI, etc.	×	×	×
Public dataset (kaggle)	Our method	NB, MLP, GBDT, RF, LR, DT	BMI, age, high BP, etc.	√	√	√

## Data Availability

The diabetes data set used to evaluate system performance in this study was from the Kaggle database: the diabetes data set from the National Institute of Diabetes and Digestive and Kidney Diseases. The data set is public and can be accessed from the following website: https://www.kaggle.com/datasets/akshaydattatraykhare/diabetes-dataset?resource=download (accessed on 9 July 2023). The heart-disease data set used to evaluate the system performance in this study was from the Kaggle database: 253,680 survey responses from Clean BRFSS 2015, primarily for the dichotomies of heart disease. The data set is public and can be accessed from the following website: https://www.kaggle.com/datasets/alexteboul/heart-disease-health-indicators-dataset (accessed on 1 January 2023).

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
