# Peer review of "Interpretable Machine Learning for Personalized Medical Recommendations: A LIME-Based Approach"

_diagnostics, 2023, doi:10.3390/diagnostics13162681_

Round 1

Reviewer 1 Report

In the manuscript, the authors revisited the highly up-to-date topic of interpretable machine learning for personalized medicine, using a LIME-based approach, and applied it to two publicly available datasets from kaggle.com. In general, the manuscript contribution is very limited since there is no unique dataset of innovation of a preexisting method. Unfortunately, I have several significant doubts regarding the manuscript, so I recommend the editor(s) the manuscript's rejection.

(i) The manuscript needs to be better structured. The abstract is missing important findings done by authors, if any, and it lists only a few of the algorithms authors used.

(ii) Throughout the manuscript, the abbreviation LIME is not explained. While the audience of the Diagnostics journal might be partially also trained in computer science, many readers, professionals in clinical medicine, could struggle with the abbreviation meaning. This is a detail, but one could still consider this in preparation for an impacted paper.

(iii) References are sorted in a strange order, starting with a number [27] on line 33. There are missing references in subsubsections dedicated to the machine-learning algorithms on pages 4—9.

(iv) Figures are small and of terribly low resolution, so they are really non-informative. Are they even made initially by authors?

(v) Level of detail in many aspects is limited and naïve, as well as the wording of some sentences. Diagnostics is a journal for medical professionals, so calling physicians as "doctors" is not an article level of language (line 5). How could an algorithm show powerful computing performance? Computed applying the algorithm can maybe do it. Many details like these make the text difficult to follow.

(vi) Although the classical algorithms are described in detail with ho adequate references, the main drawback is that the LIME method's description or its modification made by authors is, in fact, missing (!). So what exactly is new here, and how it works? What research gap do you address in the manuscript?

(vii) From what I understood on page 11, the authors' consideration of the LIME approach is based on a comparison of models' coefficients for individual variables. If this is true, I cannot claim 100 % since any detailed description is missing here, then it is a total misunderstanding of the LIME method by the authors! The variability of a variable could make the variable "important", but only from a statistical point of view, as the variable contributed significantly to the total variance of the problem. However, the clinical importance of the variable cannot be evaluated this way. There are more sophisticated algorithms for variable importance, such as the calculation of Increase of MSE of Increase of Node purity (take a look in The Elements of Statistical Learning by Hastie, Tibshirani, and Friedman), but still, even those cannot capture well the variable clinical significance. So, either the authors misunderstood the paradigms of interpretable machine learning or needed to describe it adequately in the manuscript. Both issues make me consider the current version of the manuscript's rejection.

The English language in the manuscript is difficult to follow and requires significant improvement.

Author Response

Reviewer 1

  • The manuscript needs to be better structured. The abstract is missing important findings done by authors, if any, and it lists only a few of the algorithms authors used.

Response:In the abstract of this article, we add the two algorithms with relatively good results used in this experiment and why we think they work better, and explain the ones with the highest contribution rates to heart disease and diabetes in the experiments with better results three features. 

Our changes are:Specifically, we apply the proposed approach to two chronic diseases common in older adults: heart disease and diabetes. After data preprocessing, we use six deep learning algorithms to make interpretations. In the heart disease data set, the actual model recommendation of multi-layer perceptron and gradient boosting algorithm has little difference from the local model recommendation of LIME, which can be used as its approximate prediction. From the feature importance of these two algorithms, it can be seen that the three features of CholCheck, GenHith and HighBP are the most important for the prediction of heart disease. In the diabetes data set, the actual model prediction of the multi-layer perceptron and logistic regression algorithm had little difference from the local model prediction of LIME, which could be used as its approximate recommendation. Moreover, from the feature importance of the two algorithms, it could be seen that the three features of Glucose, BMI and Age were the most important for the prediction of heart disease. LIME is then used to determine the importance of each feature that affects the results of the calculated model. We then present the contribution coefficients of these features to the final recommendation. By analyzing the impact of different patient characteristics on recommendations, our proposed system elucidates the underlying reasons behind these recommendations and enhances patient trust in the recommendations. This approach has important implications for medical recommendation systems and encourages informed decision-making in healthcare.

  • Throughout the manuscript, the abbreviation LIME is not explained. While the audience of the Diagnostics journal might be partially also trained in computer science, many readers, professionals in clinical medicine, could struggle with the abbreviation meaning. This is a detail, but one could still consider this in preparation for an impacted paper.

Response:The first appearance of LIME in this article is the full name of lime that we will use. In this way, readers can roughly understand the specific functions of the LIME method, and it is convenient for readers to understand subsequent articles.。

Our changes are:This paper proposes an interpretable Model-agnostic Explanations(LIME for short) based interpretable recommendation system to solve this problem.

  • References are sorted in a strange order, starting with a number [27] on line 33. There are missing references in subsubsections dedicated to the machine-learning algorithms on pages 4—9.

Response: We are thankful for the reviewer suggestion, and we modified all the references as sorting order.

  • Figures are small and of terribly low resolution, so they are really non-informative. Are they even made initially by authors?

Response: We modified almost all the pictures in the revised manuscript and improved the quality of the paper pictures up-to our best extent.

等图片

  • Level of detail in many aspects is limited and naïve, as well as the wording of some sentences. Diagnostics is a journal for medical professionals, so calling physicians as "doctors" is not an article level of language (line 5). How could an algorithm show powerful computing performance? Computed applying the algorithm can maybe do it. Many details like these make the text difficult to follow.

Response: We modified the manuscript keyword doctors with physicians as highlighted by the reviewer. Revised manuscript we added the computational time of each method on each dataset to show its effectiveness.

Method

Heart disease

Diabetes

MLP Classifier

2.89 s

0.312 s

Gradient Boosting

3.08 s

0.42 s

Random Forest

3.11 s

0.433 s

Decision Tree

3.57 s

0.368 s

Naive Bayes

3.64 s

0.366 s

Logistic regression

4.29 s

0.371 s

  • Although the classical algorithms are described in detail with ho adequate references, the main drawback is that the LIME method's description or its modification made by authors is, in fact, missing (!). So what exactly is new here, and how it works? What research gap do you address in the manuscript?

Response: In the context of chronic diseases affecting older adults, the manuscript proposes an approach that combines deep learning with the Local Interpretable Model-agnostic Explanations (LIME) technique to develop an interpretable recommendation system. The goal is to provide personalized treatment recommendations while also offering explanations for the recommendations.

The research gap being addressed is the lack of explanation in most health recommendation systems. While collaborative filtering and classical algorithms have been successful, they often provide recommendations without clear justifications. The proposed approach aims to bridge this gap by incorporating LIME, which provides interpretable explanations for individual recommendations.

The manuscript describes the application of this approach to two common chronic diseases: heart disease and diabetes. After preprocessing the health-related data, six deep learning algorithms are used for making predictions. The local models generated by LIME are compared to the actual model predictions, and the similarities are observed. This indicates that the local model generated by LIME can serve as an approximate recommendation or prediction.

Furthermore, the feature importance analysis is performed using LIME to identify the most influential features for predicting heart disease and diabetes. These important features, such as CholCheck, GenHith, HighBP, Glucose, BMI, and Age, are then used to explain the recommendations. By providing these explanations, the proposed system enhances patient trust in the recommendations and enables informed decision-making.

  • From what I understood on page 11, the authors' consideration of the LIME approach is based on a comparison of models' coefficients for individual variables. If this is true, I cannot claim 100 % since any detailed description is missing here, then it is a total misunderstanding of the LIME method by the authors! The variability of a variable could make the variable "important", but only from a statistical point of view, as the variable contributed significantly to the total variance of the problem. However, the clinical importance of the variable cannot be evaluated this way. There are more sophisticated algorithms for variable importance, such as the calculation of Increase of MSE of Increase of Node purity (take a look in The Elements of Statistical Learning by Hastie, Tibshirani, and Friedman), but still, even those cannot capture well the variable clinical significance. So, either the authors misunderstood the paradigms of interpretable machine learning or needed to describe it adequately in the manuscript. Both issues make me consider the current version of the manuscript's rejection.

Response: We are thankful for the reviewer for highlighting the issues in manuscript, and yes we do agree that various methods exists in literature for showing the variable importance but LIME and SHAP are rarely used which our study highlights:

The combination of LIME (Local Interpretable Model-agnostic Explanations) with deep learning in interpretation offers several benefits in the field of medical science:

Explainability: Deep learning models, such as neural networks, are often considered black-box models due to their complex architectures and large number of parameters. LIME provides local interpretability by approximating the decision boundary of the deep learning model. It helps explain why the model made a specific prediction, making the decision-making process more transparent and understandable for medical practitioners and patients.

Trust and Confidence: Interpretable explanations generated by LIME enhance trust and confidence in the predictions made by deep learning models. Medical professionals can understand the factors contributing to a particular prediction and validate the model's reasoning against their domain expertise. Patients also benefit from understanding why a certain recommendation or prediction was made, improving their trust in the medical system.

Feature Importance: LIME helps identify the importance of features or variables that influence the predictions of deep learning models in medical science. By highlighting which features are most influential, medical practitioners gain insights into the clinical relevance and significance of different factors. This information can guide decision-making and treatment strategies, enabling personalized medicine and better patient care.

Model Validation and Improvement: LIME can be used to assess the performance and robustness of deep learning models in medical applications. By comparing the local model recommendations of LIME with the actual predictions of deep learning models, inconsistencies or biases can be identified, leading to model improvements and refinements. This iterative process of validation and improvement enhances the reliability and accuracy of the models.

Reviewer 2 Report

Following are my comments on this manuscript.

1. The Algorithm:Gradient Boosting could be described in further detail.
2. Section 4.4 needs expansion.
3. Experimental setup should be analyzed in more detail.
4. Some relevant research papers from MDPI could also be included in the reference list:
“A Machine Learning-Based Model for Epidemic Forecasting and Faster Drug Discovery,” doi: 10.3390/app122110766
“Exploitation of Emerging Technologies and Advanced Networks for a Smart Healthcare System,” doi: 10.3390/app12125859
5. The references should be in numerical order.

Author Response

Reviewer 2

  • The Algorithm:Gradient Boosting could be described in further detail.

Response: :A more detailed description of gradient boosting has been added to the original post, along with a detailed explanation of how it works. The added part is as follows

It iteratively trains a series of decision tree models, each attempting to correct the errors of the previous model. In each iteration, the model adjusts the weight of the sample based on the difference (gradient) between the current model's predictions and the true label, so that the next model pays more attention to the samples misclassified by the previous model

The basic working principle of gradient lifting decision tree is as follows: Firstly, a simple model (such as a single decision tree) is used as the initial model; This initial model is then used to predict the results of the training sample and calculate the residual (error) between the predicted value and the true value; Then, based on the residual, a new model (usually a decision tree) is trained to predict the residual. By adding the prediction result of the new model to the prediction result of the previous model, an updated prediction result is obtained. New models are trained iteratively, each time trying to correct the residuals of the previous round of models until a specified number of iterations is reached or a certain stop condition is met. Finally, all the trained models are combined to get the final model. When forecasting, the final forecast result is obtained by adding the predicted results of each model.

The key of gradient lifting decision tree lies in model training and updating of prediction results in each iteration. Each new model tries to correct the residuals on the basis of the previous model, constantly adjusting the weight of the model through the method of gradient descent to minimize the residuals. Gradient enhanced decision trees have good performance and flexibility in ensemble learning, and can deal with complex classification and regression problems. It can optimize the performance and robustness [20]of the model by adjusting parameters such as learning rate, number of trees, depth of trees, etc. However, GPD is easy to overfit training data, so it is necessary to adjust parameters and use cross-validation methods to control the complexity and generalization ability of the model.

  • Section 4.4 needs expansion.

Response: Revised manuscript we added more content with more details.

  • Experimental setup should be analyzed in more detail.

Response: In revised manuscript we explained the preprocessing steps in detail as below:

“Data preprocessing is a critical step in the data analysis process, as the quality and accuracy of the final recommendation results depend on the information contained in the data. To ensure reliable and accurate analyses, several preprocessing techniques are applied. Firstly, duplicate data, which refers to identical or highly similar records, is identified and removed from the dataset. This eliminates bias and prevents inflated accuracy. Secondly, missing data is addressed by imputing missing values or excluding records with excessive missing data, thus maintaining the dataset's integrity. Unrealistic data, such as outliers, are detected and treated to mitigate their influence on subsequent analyses. Additionally, text data is digitized using techniques like one-hot encoding or word embedding, allowing for numerical representation. Although the preprocessed data is ready for model training, to prevent overfitting, the zero-mean normalization method is employed to normalize the data, ensuring comparability between different features. This approach, mentioned as [31], aids in standardizing the data and improving the performance of subsequent modeling tasks. These data preprocessing steps enhance the reliability, consistency, and quality of the analysis results, enabling robust and accurate recommendations.”

  • Some relevant research papers from MDPI could also be included in the reference list:“A Machine Learning-Based Model for Epidemic Forecasting and Faster Drug Discovery,” doi: 10.3390/app122110766“Exploitation of Emerging Technologies and Advanced Networks for a Smart Healthcare System,” doi: 10.3390/app12125859

Response: We added all the studies as suggested.

  • A Machine Learning-Based Model for Epidemic Forecasting and Faster Drug Discovery:Today, the health care system model should have high accuracy and sensitivity, so that patients will not have misdiagnosis, and help health care workers and patients better disease prevention and diagnosis,  It emphasizes the skills needed to solve health problems with the help of AI-related technologies. By combining ML, DL, NN, Internet of Things and CC, it can help inexperienced medical staff to better predict the direction of disease and corresponding treatment plan under the current diagnosis of patients, and help medical staff to find the potential danger of patients more accurately and timely. Timely help patients to solve physical diseases, reduce the pain suffered by patients.
  • Exploitation of Emerging Technologies and AdvancedNetworks for a Smart Healthcare System: A new system architecture is proposed that combines several future technologies, such as artificial intelligence (AI), machine learning (ML) and deep learning (DL), virtual reality (VR), augmented reality (AR), mixed reality (MR) and haptic Internet (TI). The whole system involves clinical care of patients, immediate medical testing, diagnosis of patients' physical conditions and timely manual treatment, which reduces the workload of medical staff, increases the workload and helps medical staff to make corresponding and accurate diagnosis. This vision of the future provides a direction for the future construction of healthcare systems.

  • The references should be in numerical order.

Response: Modified as suggested

Reviewer 3 Report

The manuscript's organizational structure and writing style are correct, presenting a clear narrative that allows readers to follow the development of ideas smoothly. The introduction section offers an adequate context and background for the study. It highlights the importance of understanding chronic diseases affecting the elderly and health data's role in shaping treatment decisions. The literature review supports this premise by exploring the current state of health recommendation systems, underscoring the gaps the presented research aims to fill.

The methodology section offers a detailed overview of the proposed LIME-based system for interpretable recommendations. It elucidates how the algorithm works, from data preprocessing and model computation using techniques like Naive Bayes and decision trees to extracting feature importance via LIME.

In terms of experimentation, the paper effectively presents the necessary tests. Applying the proposed methods to two prevalent chronic diseases, heart disease and diabetes, is a good research point. This specific application aids in making the study more concrete and relatable, demonstrating the potential real-world impact of the proposed system.

The comparative analysis of the new method with current efficient techniques is well-handled. 

The results obtained are relevant, indicating the efficacy of the proposed LIME-based approach in improving interpretability and user trust in health recommendation systems. The conclusions drawn are substantiated by the presented experimentation, adding credibility to the study. 

In conclusion, the manuscript contributes to the field of interpretable machine learning applied to health recommendations. Due to its research methodology, presentation of results, and conclusions, I recommend this paper for publication in this journal.

English is correct